# Multi-Center National Study of Genotype–Phenotype Correlation and Clinical Characteristics in Children and Young Adults with Friedreich’s Ataxia from Serbia

**DOI:** 10.3390/biomedicines13112646

**Published:** 2025-10-28

**Authors:** Gordana Kovacevic, Slobodanka Todorovic, Ivana Novakovic, Valerija Dobricic, Dusanka Savic-Pavicevic, Vedrana Milic Rasic, Marina Svetel, Milos Brkusanin, Vladislav Vukomanovic, Dragana Vucinic, Slavica Ostojic, Jovana Putnik, Ana Kosac

**Affiliations:** 1Neurology Department, Mother and Child Health Care Institute of Serbia “Dr. Vukan Cupic”, 11070 Belgrade, Serbia; slavica.ostojic@imd.org.rs; 2Faculty of Medicine, University of Belgrade, 11158 Belgrade, Serbia; danka.todorovic@gmail.com (S.T.); ivana.novakovic@med.bg.ac.rs (I.N.); vedrana.milic.rasic.npk@gmail.com (V.M.R.); marinasvetel@gmail.com (M.S.); vladislav.vukomanovic@imd.org.rs (V.V.); jovana.putnik@imd.org.rs (J.P.); ana.kosac.npk@gmail.com (A.K.); 3Clinic of Neurology and Psychiatry for Children and Youth, University of Belgrade, 11158 Belgrade, Serbia; dragana.vucinic.npk@gmail.com; 4Neurology Clinic, University Clinical Center of Serbia, 11158 Belgrade, Serbia; valerija.dobricic@gmail.com; 5Lübeck Interdisciplinary Platform for Genome Analytics (LIGA), University of Lübeck, 23562 Lübeck, Germany; 6Center for Human Molecular Genetics, Faculty of Biology, University of Belgrade, 11158 Belgrade, Serbia; duska@bio.bg.ac.rs (D.S.-P.); milos.brkusanin@bio.bg.ac.rs (M.B.); 7Cardiology Department, Mother and Child Health Care Institute of Serbia “Dr. Vukan Cupic”, 11070 Belgrade, Serbia; 8Nephrology Department, Mother and Child Health Care Institute of Serbia “Dr. Vukan Cupic”, 11070 Belgrade, Serbia

**Keywords:** Friedreich’s ataxia, GAA repeat, genotype–phenotype correlation, cardiomyopathy, nephrotic syndrome

## Abstract

**Background/Objectives:** Friedreich’s ataxia (FA) is a rare neurodegenerative disorder caused by GAA repeat expansions in the *FXN* gene. While well-studied in larger populations, data from Southeastern Europe are limited. This study aimed to characterize the clinical and genetic features of FA in a Serbian cohort and explore genotype–phenotype correlations. **Methods:** A multi-center, retrospective analysis was conducted on 30 genetically confirmed FA patients. Clinical assessments included neurological, cardiological, and metabolic evaluations. GAA repeat sizes were determined in 26 patients, and correlations with clinical features were analyzed. **Results:** The mean age at disease onset was 9.0 ± 3.0 years, with ataxia as the initial symptom in 80% of patients. Hypertrophic cardiomyopathy was present in 73.3%, and 43.3% of patients lost ambulation within 1.5 to 15 years after symptom onset. Two patients developed diabetes, and two were diagnosed with nephrotic syndrome. Genetic analysis revealed an average GAA1 repeat length of 805 and GAA2 of 1024 alleles. Larger GAA1 expansions were associated with extensor plantar responses, while longer GAA2 repeats correlated with impaired vibration sense. Disease duration was strongly linked to multiple neurological signs and loss of ambulation. No significant correlation was found between GAA repeat length and age at onset. **Conclusions:** This study provides the first genotype–phenotype analysis of FA in Serbia, confirming known patterns and revealing new comorbidities, such as nephrotic syndrome. GAA repeat length influences some clinical features but does not fully predict disease onset or progression, indicating the need for broader genetic and environmental studies.

## 1. Introduction

Friedreich’s ataxia (FA) is the most prevalent inherited ataxia in individuals of Caucasian descent, with a prevalence of approximately 1 in 30,000 to 1 in 50,000 individuals [1,2]. This progressive neurodegenerative disorder is characterized by limb and gait ataxia, areflexia, dysarthria, and multisystem involvement, including hypertrophic cardiomyopathy and diabetes mellitus [3,4,5,6,7,8,9,10,11]. FA is caused by a guanine–adenine–adenine (GAA) repeat expansion in the frataxin (*FXN*) gene, with 96–98% of patients being homozygous for the GAA expansion, while 2–4% are compound heterozygotes with one allele carrying the GAA expansion and the other harboring a point mutation or deletion [12,13,14,15]. Normal alleles typically contain between 6 and 27 uninterrupted GAA repeats, while expanded alleles exhibit a range of 120 to 1700 repeats [6,12,13,14,16].

Additionally, some expanded alleles contain small interruptions in the terminal regions of the GAA repeat tract, which have been associated with milder or atypical disease phenotypes and later disease onset [17]. A recent study identified a single-nucleotide polymorphism in the SIRT6 (Sirtuin 6) gene, resulting in the S46N protein change, as a modifier of disease severity in FA, suggesting the potential involvement of genetic modifiers [18].

While FA has been extensively studied in Western populations, data from Southeast Europe remain scarce. Furthermore, while the relationship between GAA repeat length and clinical manifestations has been well established in adult cohorts [6,19,20,21,22,23,24,25,26,27,28,29,30], genotype–phenotype correlations in pediatric cases, particularly in the context of early disease onset, remain underexplored.

This study aims to address this gap by investigating the clinical features, comorbidities, and genotype–phenotype associations in a cohort of Serbian pediatric and young adult FA patients. By analyzing GAA repeat lengths and clinical features, we aim to further understand the relationship between genotype and disease progression in early-onset cases. Additionally, we identify comorbidities such as nephrotic syndrome, which has rarely been reported in FA, providing new insights into the disease’s systemic impact.

## 2. Materials and Methods

### 2.1. Study Design and Participants

This multi-center, retrospective study included 30 patients with genetically confirmed Friedreich’s ataxia, originating from 25 non-consanguineous families. Initially, 34 patients were considered based on clinical suspicion of FRDA. However, four patients were excluded from the analysis: two patients had died before genetic testing, one was not genetically tested, and one was asymptomatic despite a positive genetic result. These patients were therefore excluded from both the genotype–phenotype analysis and the clinical evaluation (Figure 1). Additionally, of the 30 genetically confirmed patients, GAA repeat sizing was unsuccessful in four cases due to suboptimal DNA quality, likely resulting from extraction from long-stored blood samples. It should be noted that the initial genetic diagnostic reports did not include repeat size information; GAA sizing was performed later as part of the present study. Participants were evaluated at three tertiary care centers in Belgrade: The Clinic for Neurology and Psychiatry for Children and Youth, the Institute for Mother and Child Health Care of Serbia, and the Neurology Clinic at the Clinical Center of Serbia. Ethical approval was obtained from the Institutional Ethics Committees of all participating centers, in accordance with the Declaration of Helsinki. Written informed consent was obtained from the parents or legal guardians of all patients.

### 2.2. Clinical Assessment

Clinical data were collected through a review of medical records and standardized questionnaires completed by parents. Patients were diagnosed according to Harding’s clinical criteria, which include early onset (before 25 years), progressive gait and limb ataxia, and areflexia. Additional features, such as dysarthria, extensor plantar responses, and sensory neuropathy—defined by reduced or absent sensory nerve action potential (SNAP) amplitudes, with motor conduction velocities in the arms exceeding 40 m/s—were also considered [5]. The patients were classified according to the age of onset (AOO) into early (0–7 years), typical (8–14 years), intermediate (15–24 years), and late onset (>24 years) [30].

All patients underwent detailed neurological examinations conducted by experienced neurologists at the three clinical centers involved in the study. Vibration perception was assessed using a 128 Hz tuning fork, a widely used method for evaluating vibration sense. This method relies on the patient’s ability to perceive and accurately report the sensation, which can introduce subjectivity, particularly in pediatric populations.

Cardiovascular evaluations, including electrocardiography (ECG), were performed in all patients, while transthoracic echocardiography (TTE) was conducted in all except one. Fasting serum glucose was assessed in all patients, and seven individuals (21%) underwent an oral glucose tolerance test (OGTT).

### 2.3. Genomic Analysis

Genomic DNA was extracted from blood samples using the Qiagen Blood Mini Kit (Qiagen, Hilden, Germany) according to the manufacturer’s instructions, and the *FXN* gene was analyzed for GAA repeat expansions using long-range repeat-primed polymerase chain reaction (PCR). The sizes of the GAA repeats were determined by gel electrophoresis, and GAA allele lengths were calculated using a standard formula. The sizes of the smaller and larger alleles were designated as GAA1 and GAA2, respectively.

### 2.4. Statistical Analysis

Statistical analyses were performed using R Statistical Software (version 2.8.1) (R Foundation for Statistical Computing, Vienna, Austria) [31]. Normality of data distribution was assessed using the Kolmogorov–Smirnov and Shapiro–Wilk tests. Depending on the distribution, comparative analyses were conducted using the Wilcoxon rank sum test, the Wilcoxon test with continuity correction, or the Exact Wilcoxon rank sum test. Spearman’s rank correlation coefficients were used to examine relationships between GAA repeat lengths and clinical features. Results are presented as mean ± standard deviation (SD), range, or frequencies (%). A *p*-value < 0.05 was considered statistically significant.

## 3. Results

### 3.1. Patient Demographics and Clinical Characteristics

The study cohort comprised 30 patients (16 females, 14 males) with a mean age at disease onset of 9.0 ± 3.0 years. Most patients (53.3%) exhibited typical onset between 8 and 14 years, while 43.3% presented with early onset. The mean age at clinical assessment was 16.9 ± 4.0 years. Ataxia was the initial symptom in 80% of patients, followed by scoliosis in 20% [32,33].

Hypertrophic cardiomyopathy (HCM) was observed in 73.3% of patients, while ECG abnormalities were noted in 90%, with T wave inversion (60%) and left ventricular hypertrophy (23.3%) being the most common findings. Diabetes mellitus was diagnosed in two patients (6.7%), manifesting 6 and 15 years after disease onset, respectively. Additionally, two patients (6.7%) had nephrotic syndrome as a comorbidity. All patients met Harding’s obligatory criteria for typical FA; however, secondary features indicated atypical phenotypes in 15 patients (50%).

A summary of clinical features is presented in Table 1.

The mean disease duration was 8 ± 4 years. Among the patients, 13 (43.3%) lost the ability to walk independently within 1.5 to 15 years after symptom onset. Notably, 8 patients (26.7%) became wheelchair-bound between 2.5 and 14 years (mean 8.25 ± 4 years, median 8.5 years).

### 3.2. Genetic Analysis

The mean size of the shorter expanded GAA allele (GAA1) was 805 ± 177 repeats (range 329–1060, median 851), while the mean size of the longer allele (GAA2) was 1024 ± 118 repeats (range 824–1294, median 1058). Forty percent of the alleles fell within 800–1000 repeat ranges [32].

The distributions of GAA1 and GAA2 expansion sizes are illustrated in Figure 2 and Figure 3, respectively.

### 3.3. Genotype–Phenotype Correlations

To explore genotype–phenotype correlations, we compared the total GAA1 and GAA2 lengths, as well as age at onset and disease duration, between patients with and without specific clinical features (Appendix A).

A correlation analysis revealed that a larger GAA1 allele size was significantly associated with the presence of an extensor plantar response (*p* = 0.024). Additionally, a larger GAA2 allele size was associated with impaired vibration sense (*p* = 0.025).

Neither the size of GAA1 nor GAA2 repeat lengths was a significant predictor of AOO in our cohort (*p* = 0.552 and *p* = 0.953, respectively). However, AOO demonstrated a significant inverse correlation with the time to loss of independent ambulation (rho = −0.143, *p* = 0.006), suggesting that later onset may be associated with more rapid disease progression.

Disease duration was significantly longer in patients with dysarthria (*p* = 0.003), upper limb areflexia (*p* = 0.015), nystagmus (*p* = 0.036), foot deformities (*p* = 0.031), ECG abnormalities (*p* = 0.032), and loss of independent walking (*p* = 0.002). Detailed correlation data are provided in Table 2.

### 3.4. Intrafamilial Variability

Among the three non-consanguineous families with multiple affected siblings, the GAA1 repeat size difference ranged from 10 to 250 repeats (Table 3). In the family with the largest GAA1 size difference (250 repeats), the clinical expression of the disease was notably more variable, with siblings experiencing different ages of onset and disease severity. In contrast, the siblings with a smaller GAA1 size difference (10 to 40 repeats) showed a more similar disease presentation. Notably, in one family, identical twin sisters (Patients I2 and I3) developed symptoms six years later than their older brother, despite nearly identical clinical features. A single difference was observed between the twins: cardiomyopathy appeared later in Patient I3.

## 4. Discussion

This study evaluated the clinical and genetic features of Friedreich’s ataxia (FA) in a cohort of 30 children and young adults from Serbia. The mean age at onset in our cohort was consistent with previous studies, with ataxia being the most common initial symptom, followed by scoliosis [5,6,10,22,23,24,25,26,34].

Hypertrophic cardiomyopathy (HCM), confirmed by transthoracic echocardiography, was observed in 73.3% of patients, with an equal proportion exhibiting electrocardiographic abnormalities. These findings are in line with prior studies identifying HCM as a hallmark feature of FA, present in over two-thirds of patients and often leading to significant cardiac dysfunction [6,35,36,37]. Electrocardiographic abnormalities—such as ventricular repolarization disturbances and right axis deviation—are frequently reported, affecting up to 90% of individuals [38]. These data underscore the importance of routine cardiovascular surveillance in this population, as timely identification of cardiac involvement can significantly impact long-term outcomes.

Diabetes mellitus (DM) was diagnosed in two patients (6.7%) within our cohort, aligning with previously reported prevalence rates of 6–9% in individuals with Friedreich’s ataxia [39,40]. Up to 30% of individuals with FA may develop impaired glucose tolerance, likely due to mitochondrial dysfunction in pancreatic β-cells [41,42]. Further research into the molecular mechanisms underlying this comorbidity may inform targeted therapeutic strategies.

The co-occurrence of nephrotic syndrome (NS) in two patients is noteworthy. The first patient is a female child diagnosed with steroid-sensitive nephrotic syndrome (SSNS) at 18 months of age. As the disease was characterized by only two relapses over the subsequent 18 months, no kidney biopsy was indicated. To date, remission of nephrotic syndrome and stable renal function have been maintained. However, at the age of six, she developed symptoms consistent with Friedreich’s ataxia.

The second patient is a young male who first exhibited symptoms of FA at the age of six. At 17 years old, he presented with periorbital and generalized edema. Based on clinical and laboratory findings, a diagnosis of nephrotic syndrome was established, and corticosteroid therapy was initiated. Due to oliguric acute kidney injury, methylprednisolone pulses were administered. Renal function normalized after the fifth pulse, and urinary remission was achieved in the third week of corticosteroid therapy. A kidney biopsy was performed, revealing minimal change disease (MCD). Due to frequent relapses, he was subsequently treated with cyclosporine and mycophenolate mofetil over the next three years. At 20 years of age, upon transfer to adult nephrology care, stable remission was maintained; however, glomerular filtration rate began to decline.

To the best of our knowledge, an association between FA and NS has been documented in only four patients across multiple families [43,44]. Watters et al. described a family with four siblings, three of whom were diagnosed with FA, and two developed NS with kidney biopsy findings consistent with minimal change disease. One of these patients had SSNS with steroid dependency and died at the age of 20; the second achieved stable remission. The authors proposed that the co-occurrence of both conditions in two siblings was unlikely to be coincidental, suggesting a shared, albeit unidentified, immunological abnormality [43]. Similarly, Shinnick et al. reported two additional cases of FA associated with SSNS. Both patients achieved remission, and one of them demonstrated neurological improvement following corticosteroid therapy [44].

A detailed comparison of our cases with those previously reported, including the age of NS onset, type of NS, and histopathological findings, is provided in Table 4.

Emerging evidence implicates increased production of reactive oxygen species (ROS) and impaired antioxidant defense systems in the pathogenesis of proteinuria in NS [45]. Conversely, frataxin—a mitochondrial protein deficient in FA—has been shown to play a role in modulating oxidative stress by reducing ROS levels [9,46]. These findings raise the possibility that the co-occurrence of FA and NS may not be incidental, but rather reflective of a shared pathophysiological process involving oxidative stress and tissue injury. Further investigations, particularly those evaluating frataxin expression and mitochondrial function in podocytes, are warranted to elucidate this potential mechanistic link.

Genotype–phenotype correlations revealed significant associations between GAA1 repeat length and extensor plantar responses, and GAA2 repeat length with impaired vibration sense. These findings align with previous studies linking GAA repeat length to specific clinical manifestations, such as peripheral neuropathy, areflexia, pes cavus, and scoliosis [5,6,19,21].

No significant associations were found between GAA repeat size and the presence of cardiomyopathy, abnormal ECG, or DM. Although some studies suggest longer GAA repeats may be linked to cardiac involvement [5,19,25], others emphasize disease duration and earlier onset as stronger predictors [20,24,28]. Regarding diabetes mellitus, while McCormick et al. found a significant relationship with GAA size [39], other studies have not [6,20], and some propose age at onset as a more relevant factor [20].

We did not find a significant correlation between GAA repeat size and age at onset. This finding does not align with most existing literature suggesting an inverse relationship [10,19,20,21,22,23,24,30], possibly due to the relatively narrow age at onset range (6–15 years) in our cohort, which may not capture the full spectrum of disease variability.

The GAA1 expansion has been established as a major determinant of disease severity, and explains ~50% of the variability in age at onset [6,19,26,30]. The lower threshold of GAA1 repeats associated with later onset varies between studies (500–800 repeats) [6,19], and residual *FXN* transcriptional activity from shorter alleles may underlie milder phenotypes [6,19,22,23,26,27].

In our cohort, disease duration was significantly associated with several clinical features, including dysarthria, upper limb areflexia, foot deformities, ECG abnormalities, and loss of independent walking. These findings are consistent with previous studies, which demonstrated that longer disease duration is often associated with a higher burden of neurological and systemic complications [3,5,6,26,30,47,48].

We also observed that 43.3% of patients lost independent ambulation within 1.5 to 15 years following disease onset, while 26.7% became wheelchair-dependent within 2.5 to 14 years. These findings are consistent with previous reports indicating that loss of ambulation generally occurs within 10 to 20 years of disease onset, although with considerable variability, influenced by factors such as genotype, age at onset, and other clinical parameters [6,20,30,47,48,49].

Our findings indicated that a later age at disease onset was associated with a more rapid progression to loss of independent ambulation. Although statistically significant, the effect size was weak, indicating limited clinical relevance. Such a low correlation coefficient suggests that age at onset, while potentially contributing to disease progression, is likely influenced by multiple other factors. Therefore, this finding should be interpreted with caution and further investigated in larger and more diverse cohorts. However, this unexpected result may be influenced by the limited sample size in our cohort or inaccuracies in reported onset ages, which could have affected the strength and direction of the observed correlation.

Finally, notable intrafamilial variability was observed despite identical GAA expansions among siblings. This supports prior observations of heterogeneity in FA expression in siblings [50,51] and suggests that additional research into genetic modifiers—such as mitochondrial haplogroups, modifier genes, and epigenetic mechanisms—is crucial to understanding the full spectrum of disease expression in FA.

## 5. Limitation

While this study provides valuable insights into the clinical and genetic characteristics of Friedreich’s ataxia in a Serbian cohort, there are several limitations. The primary limitation of our study is the relatively small sample size (N= 26), which may have reduced the statistical power of our correlation analysis. Additionally, the narrow age range at onset (6 to 15 years) may have constrained the variability needed to detect a significant correlation between GAA repeat size and age of onset. Furthermore, only a small number of alleles exhibited fewer than 600 or more than 1100 repeats, further limiting the ability to establish a robust relationship between GAA repeat size and disease progression.

Another important limitation is the absence of standardized clinical rating scales commonly used in FRDA research, such as the Scale for the Assessment and Rating of Ataxia (SARA), the modified Friedreich’s Ataxia Rating Scale (mFARS), and the International Ataxia Rating Scale (INAS). While baseline neurological assessments provided useful clinical data, the lack of these validated scales limits the comparability of our findings with other studies and restricts the ability to objectively monitor disease progression or treatment response. Recognizing this, we plan to incorporate internationally accepted clinical rating tools, such as mFARS and SARA, into future prospective studies.

Finally, the assessment of vibration perception using a 128 Hz tuning fork, though this tool is commonly used for clinical screening, is subjective and dependent on the patient’s ability to accurately report sensation. This method may be less reliable in younger patients or those with cognitive difficulties, potentially introducing variability in our findings. More objective methods, such as biothesiometry or quantitative sensory testing (QST), would provide more consistent and reliable measurements of vibration perception, and we intend to utilize these tools in future investigations.

## 6. Conclusions and Further Directions

This study represents the first comprehensive clinical and genetic investigation of Friedreich’s ataxia in a pediatric and young adult cohort from Serbia, reinforcing the multisystemic nature of the disease.

Although specific genotype–phenotype correlations, such as the relationship between GAA repeat size and neurological signs, were confirmed, our findings underscore the limitations of using GAA repeat length as a sole predictor of disease onset or progression. The absence of a significant correlation between GAA repeat size and age at onset suggests the potential influence of additional genetic, epigenetic, or environmental factors in determining disease severity.

Moreover, while significant associations between disease duration and clinical features were identified, the cross-sectional nature of the study limits the ability to assess long-term disease progression. Longitudinal studies are necessary for a deeper understanding of FA’s natural history and trajectory.

The identification of nephrotic syndrome as a potential comorbidity, along with high rates of cardiac and metabolic involvement, highlights the need for multidisciplinary surveillance and reinforces the systemic nature of FA.

Future research should aim to elucidate the molecular mechanisms responsible for intrafamilial phenotypic variability. Investigating mitochondrial haplogroups, tissue-specific mosaicism, and other genetic and epigenetic modifiers may provide further insights into the diverse clinical manifestations observed in individuals with the same pathogenic variant. Additionally, large-scale studies involving diverse populations are essential to identify broader epidemiological patterns and to enhance clinical management strategies for Friedreich’s ataxia. Such efforts will contribute to more accurate predictions of disease progression and support the development of personalized therapeutic approaches, ultimately improving outcomes for patients with FA.

## Figures and Tables

**Figure 1 biomedicines-13-02646-f001:**
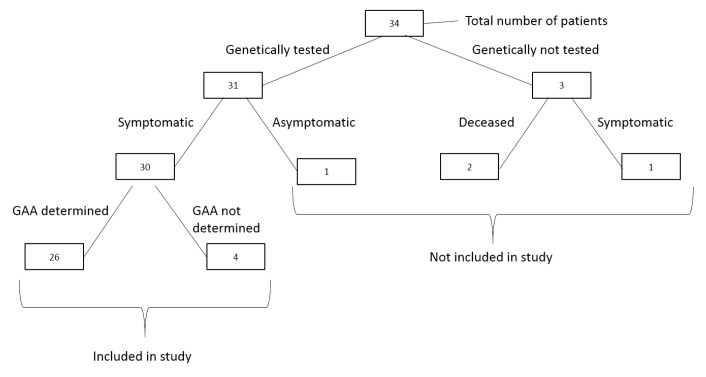
Overview of patient inclusion, genetic testing status, and determination of GAA repeat number.

**Figure 2 biomedicines-13-02646-f002:**
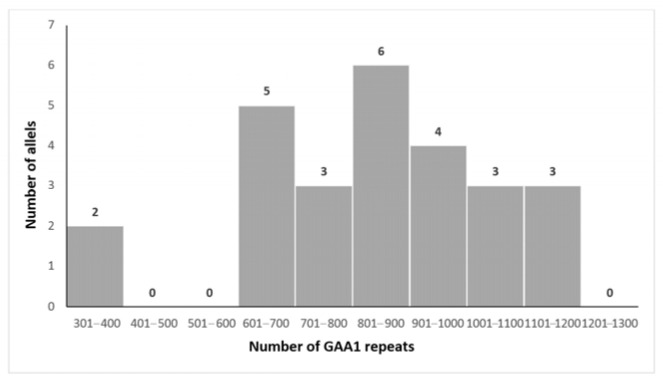
Distribution of the size of the GAA1 expansions in our patients.

**Figure 3 biomedicines-13-02646-f003:**
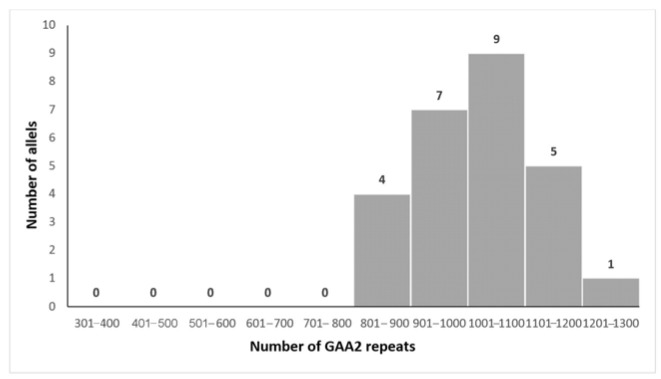
Distribution of the size of the GAA2 expansions in our patients.

**Table 1 biomedicines-13-02646-t001:** Clinical characteristics of 30 patients with ataxia.

Clinical Characteristics	Number of Patients	Percentage
1. Neurological		
Ataxia	30/30	100%
Romberg sign	30/30	100%
Lower limb areflexia	30/30	100%
Upper limb areflexia	28/30	93.3%
Extensor plantar response	20/28	71.4%
Dysarthria	16/30	53.3%
Nystagmus	10/30	33.3%
2. Orthopedic		
Scoliosis	26/30	86.7%
Foot deformities	23/28	76.7%
3. Cardiometabolic		
Cardiomyopathy	22/29	73.7%
Diabetes mellitus	2/30	6.7%
4. Comorbidity		
Nephrotic syndrome	2/30	6.7%

The number in the denominator represents the number of patients upon whom information was available to assess the corresponding parameter.

**Table 2 biomedicines-13-02646-t002:** Matrix of Spearman’s rank-correlations for age at onset (years), disease duration to loss of independent walk (years), disease duration to becoming wheelchair-bound, and GAA expansion sizes.

	GAA1		GAA2		GAA1-GAA2 Total		Age at Onset	
	Rho	*p*-Value	Rho	*p*-Value	Rho	*p*-Value	Rho	*p*-Value
Age at onset	−0.122	0.552	−0.112	0.953	−0.078	0.706	n.a.	n.a
Disease duration to loss of independent walking	−0.143	0.658	−0.046	0.886	−0.061	0.851	−0.718	**0.006**
Disease duration to becoming wheelchair-bound	0.286	0.556	0.286	0.556	0.286	0.556	−0.642	0.086

Rho: Spearman’s rank correlation coefficient; bold: statistically significant difference. GAA1 and GAA2: the number of GAA repeats in the smaller and the larger *FXN* allele, respectively. GAA-total: sum of GAA1 and GAA2 alleles.

**Table 3 biomedicines-13-02646-t003:** Some clinical characteristics and the number of GAA1 and GAA2 repeats in families with multiple affected siblings.

	Family I			Family II		Family III		
Characteristic	I-1	I-2	I-3	II-1	II-2	III-1	III-2	III-3
GAA1	850	890	880	876	858	659	919	-
GAA2	1070	1071	1066	943	1014	1168	1148	-
Age at onset (years)	8	14	14	8	6	13	12	15
Age at examination	16	21	21	18	11	18	23	21
Independent walk *	- (16)	- (21)	- (21)	+	+	+	- (19)	+
Positive Babinski sign	+	+	+	-	-	-	+	+
Pes cavus	+	+	+	+	+	-	+	-
Scoliosis	+	+	+	+	+	+	+	-
Cardiomyopathy ^$^	+ (10)	+ (15)	+ (19)	+ (15)	-	-	-	+ (18)
Dysarthria	+	+	+	-	-	-	+	-

GAA1 and GAA2: the number of GAA repeats in the smaller and the larger *FXN* allele, respectively; “-”: absent; “+”: present; *: values in parentheses represent age at wheelchair confinement; $: values in parentheses represent age at diagnosis.

**Table 4 biomedicines-13-02646-t004:** Nephrotic syndrome in patients with Friedreich’s ataxia: reported vs. our cases.

Author	Patient’s Number	Age at FA Onset	Number of GAA1/ GAA2	Age at NS Onset	Clinical Type of NS	Pathohistological Changes	Course	Outcome
Watters, et al. Can J Neurol Sci, 1981 [43]	1	6 years	NA	5 years	SSNS	MCD	Frequent relapses	Steroid-dependent NS, died at 20
	2	5.5 years	NA	12 years	SSNS	MCD	Several relapses	Stable remission
Shinnick, et al. BMC Neurol. 2016 [44]	3	10 years	650/850	2 years	SSNS	Biopsy not indicated	Several relapses	Stable remission
	4	7 years	650/1000	5 years	SSNS	Biopsy not indicated	Several relapses	Stable remission
Our study	5	6 years	329/824	18 months	SSNS	Biopsy not indicated	A few relapses	Stable remission
	6	6 years	NA	17 years	SSNS	MCD	Frequent relapses	Remission, but GFR has declined at 20

Legend: FA—Friedreich ataxia; GAA1, GAA2—number of GAA1 and GAA2 repeats, respectively; NS—nephrotic syndrome; SSNS—steroid sensitive nephrotic syndrome; MCD—minimal changes disease; GFR—glomerular filtration rate.

## Data Availability

The original contributions presented in this study are included in the article. Further inquiries can be directed to the corresponding author.

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
