# Peer review of "Multi-Center National Study of Genotype–Phenotype Correlation and Clinical Characteristics in Children and Young Adults with Friedreich’s Ataxia from Serbia"

_biomedicines, 2025, doi:10.3390/biomedicines13112646_

Round 1
Reviewer 1 Report
Comments and Suggestions for Authors
I would like to congratulate the authors for their valuable effort in defining for the first time a cohort of pediatric and adolescent patients with Friedreich's ataxia (FRDA) in Serbia. The establishment of national cohorts is important for improving recognition, care, and future research on rare diseases.
Summary
This manuscript describes a cohort of patients with Friedreich's ataxia and examines genotype-phenotype correlations. It provides useful baseline data, but remains descriptive and is limited by several methodological and reporting gaps. Below I list points that I believe should be addressed to improve understandability, reproducibility, and clinical interpretability.
- Use of clinical rating scales and standardization of assessments
- The manuscript does not report standardized clinical scales commonly used in FRDA research (e.g. SARA, mFARS, INAS). Please indicate whether these scales were used; if not, explicitly acknowledge this limitation.
- If assessments were based on baseline neurological examination only, please explain plans to implement internationally accepted scales for prospective follow-up, so that future data are comparable with other cohorts and suitable for monitoring treatment.
- Discrepancies in patient numbers and genetic confirmation
- Patient numbers are inconsistent in the abstract and main text (e.g. “GAA repeat sizes were determined in 26 patients” vs. “study included 30 patients with genetically confirmed FRDA… 34 affected members… 2 patients died, 1 not genetically tested, 1 asymptomatic but genetically positive”). I recommend presenting these numbers in one clear table that shows: total number tested, genetically confirmed, genetically not tested, deceased, asymptomatic carriers and number with available GAA size for both alleles.
- Patients without molecular confirmation (DNA testing) should not be counted as genetically confirmed cases. If clinically suspected but untested patients were included, they should be clearly identified as such or excluded from genotype-phenotype analyses. From the literature and my own experience, meeting Harding's clinical criteria does not guarantee a diagnosis. It may represent a number of other autosomal recessive ataxias – and even some autosomal dominant conditions due to anticipation.
- Missing repeat GAA size determination (four patients)
- Please clarify which four patients had GAA size not determined and explicitly state the technical reasons (e.g. poor DNA quality, test failure, test not performed). Indicate whether these are the same cases identified as "not genetically tested/deceased/asymptomatic".
- Methods for testing vibration perception (and other clinical measures)
- The article reports an association between GAA2 allele size and impaired vibration perception, but the method used to assess vibration perception is not described. Please provide the exact method (e.g., 128 Hz tuner, 64 Hz Rydel-Seiffer tuner, vibration threshold biothesiometer/neurothesiometer, or QST). If a tuner was used, discuss its subjectivity and limitations—especially in children—and consider qualifying this finding if objective VPT/QST data are not available.
- References and previous data presentations
- There appears to be an older abstract/presentation (ref. 32). Please clarify whether this manuscript contains new data or overlaps with a conference abstract, and edit references accordingly.
Minor suggestion for the future:
- A highly debatable finding is the lack of correlation between the number of GAA repeats and disease manifestation. However, it is possible that with future refinement of the patients’ data and more detailed longitudinal follow‑up of the entire cohort this finding may be revised. It is generally accepted that a longer GAA1 expansion predicts a faster/more severe disease course (e.g., Reetz K, Dogan I, Costa AS, et al. Biological and clinical characteristics of the European Friedreich's Ataxia Consortium for Translational Studies (EFACTS) cohort: a cross‑sectional analysis of baseline data. Lancet Neurol. 2015 Feb;14(2):174‑82. doi: 10.1016/S1474-4422(14)70321-7. Epub 2015 Jan 5. PMID: 25566998).
- Consider collaborating with adult neurologists to capture adult-onset and late-diagnosis cases; as you note, some adults report childhood symptoms only after detailed follow-up interviews.
Conclusion and Recommendations
This manuscript documents a valuable national effort and, after revision, could be suitable for publication (see points 1–5 above). After these revisions, I would support publication as a descriptive baseline report with the clear caveat that prospective standardized assessments and further genetic/phenotypic characterization will be needed.
Reviewer 2 Report
Comments and Suggestions for Authors
My comments are present in the attached file.

Reviewer 3 Report
Comments and Suggestions for Authors
This manuscript presents the results of a comprehensive clinical and genetic study of Friedreich's ataxia in a pediatric and young adult cohort from Serbia. Generally, this manuscript is written precisely and seems educational for the rare disease of Friedreich’s ataxia. Although the author described the narrow age range at onset (6 to 15 years) as a limitation, this seems to be a strength in this study and is helpful for clinical practice in pediatric neurology.
These need to be a few minor revisions.
- We already have several reviews of Freidreich’s ataxia, and many results in this study support the previous ones. In this study, the identification of nephrotic syndrome as a potential comorbidity is an important point. The author should describe previous cases of nephrotic syndrome (NS) more precisely, for example, the age of onset of NS, pathologic findings, or type of NS, and review them in a table to compare them with those in this study.
- The author compared mean GAA1 and GAA2 lengths, not the total of GAA, GAA1 and GAA2, length to explore genotype–phenotype correlations. Did the author examine the relationship between the total of GAA length and phenotypes?
